# Association between the Area of the Highest Flank Temperature and Concentrations of Reproductive Hormones during Pregnancy in Polish Konik Horses—A Preliminary Study

**DOI:** 10.3390/ani11061517

**Published:** 2021-05-23

**Authors:** Małgorzata Maśko, Łukasz Zdrojkowski, Małgorzata Wierzbicka, Małgorzata Domino

**Affiliations:** 1Department of Animal Breeding, Institute of Animal Science, Warsaw University of Life Sciences, 02-787 Warsaw, Poland; malgorzata_masko@sggw.edu.pl; 2Department of Large Animal Diseases and Clinic, Institute of Veterinary Medicine, Warsaw University of Life Sciences, 02-787 Warsaw, Poland; lukasz_zdrojkowski@sggw.edu.pl (Ł.Z.); malgorzata_wierzbicka@sggw.edu.pl (M.W.)

**Keywords:** Konik Polski, gestation, thermography, progesterone, estrogens, relaxin

## Abstract

**Simple Summary:**

Thermographic imaging is a potentially useful tool in pregnancy detection in equids, especially native or wildlife breeds. In late gestation, the differences in temperature of the flank area between pregnant and non-pregnant mares were suspected to be caused by hormonal effects on regional blood flow. This preliminary study aimed to assess changes in thermal features of the abdomen lateral surface and concentrations of reproductive hormones in Polish native pregnant mares. The study was carried out on pregnant and non-pregnant Polish native Konik Polski mares, and both the hormonal profile and the thermal pattern of mares’ abdomens were examined monthly. The features from thermal images in rainbow HC and gray palette were extracted to determine relations with a hormonal profile dependent across the months of pregnancy. The new feature of a thermal image in the gray palette, the area with the highest temperatures (Area of Tmax), was found to be significantly related to progesterone and relaxin blood concentration. The standard features of the thermal image did not show any similar associations over consecutive months of pregnancy. In this preliminary study, a new approach to thermographic imaging analysis based on the Area of Tmax calculation was proposed. However, further research is needed to better understand the relation between the Area of Tmax and concentrations of reproductive hormones, especially including an experiment with at least daily blood sampling and the hair coat lengths examination.

**Abstract:**

Determination of the pregnancy status is one of the most important factors for effective pregnancy management. Knowledge of the stage of pregnancy is important to interpret many of the reproductive hormones’ concentrations, including progesterone (P4), estrone sulfate (E1S), 17-ß estradiol (E2), and relaxin (REL). However, it is limited in wildlife or captive equids that cannot be handled. Reproductive hormones affect regional blood flow, the proliferation of tissues, and local metabolism intensity. Therefore, this preliminary study aimed to assess changes in thermal features of the abdomen lateral surface and concentrations of reproductive hormones in Polish native pregnant mares. The study was carried out on 14 non-pregnant and 26 pregnant Polish Konik mares during eleven months of pregnancy. Infrared thermography was conducted to image the lateral surface of mares’ abdomen (Px1) and flank area (Px2); P4, E1S, E2, and REL concentrations in serum were also determined. The evidence of the association between the area with the highest temperatures (Area of Tmax) and serum concentrations of P4 (the slope = 1.373; *p* = 0.9245) and REL (the slope = 1.342; *p* = 0.4324) were noted dependent across months of pregnancy. Measures of superficial body temperatures were found to change monthly, similarly to ambient temperatures, with no evidence of coincidence with changes in reproductive hormone concentrations. Individual thermal characteristics of the lateral surface of the abdomen differed between pregnant and non-pregnant mares in other periods. Differences in maximal and average temperature and Area of Tmax were observed from the sixth month of pregnancy, and those in minimal temperature were observed from the eighth month.

## 1. Introduction

Thermographic imaging has been reported as a potentially useful tool in pregnancy detection in equids since 2009, particularly in the field of late gestation assessment [1,2,3]. Bowers et al. described significant differences in temperature of the flank area between pregnant and non-pregnant mares beginning at 292 days of gestation. Authors assumed that the differences could be caused by hormonal effects on regional blood flow, the proliferation of tissues, metabolic and/or hormonal interactions, protein synthesis activity, or any number of associations with pregnancy that relate to the placenta and the fetus [1]. All these processes require mares to expend a large amount of energy, which produces metabolic heat in tissues leading to an increase in body temperature [4]. The radiant energy emitted by the body surface may be detected and registered using infrared thermography [5]. Both local alterations in metabolic processes [6,7] and blood flow [8] have been previously described in horses as strongly conditioning the surface body temperature. Therefore, it can be hypothesized that the reproductive hormone concentrations and superficial body temperature may be related. It also should be kept in mind that the temperature measured from the body surface is related not only to the internal conditions but also to the thermal properties of the skin and hair coat [9] and the thermal gradient between the skin surface and the environment [10,11].

Determination of the pregnancy status is one of the most important factors for effective pregnancy management. The most valuable method in pregnancy stage diagnosis in mares is transrectal ultrasonography [12]. However, this method cannot be used in wild horses; thus, there is the need for finding a method that does not require direct contact with an animal [1]. Knowledge of stage of pregnancy is important to interpret many of the reproductive hormones’ concentrations, including progesterone (P4), estrone sulfate (E1S), 17-ß estradiol (E2), and relaxin (REL) [13,14]. Various progesterone (pregn-4-ene,3,20-dione and its metabolite 5α-dihydroprogesterone (DHP)) and estrogens (estrone, 17-beta estradiol, equilin, and equilenin) concentrations may reflect placental and fetal health [13,15,16], whereas relaxin may indicate the impaired placental function [17]. Pregnant mares’ P4 concentrations gradually increase in the second and third trimesters up to the concentration of 2–12 ng/mL. In the last week before parturition, P4 concentrations increase rapidly, forming a progesterone peak. After 305 days of gestation, an increase in progesterone concentration could be a part of a physiological sequence of hormonal changes preceding the delivery [18,19]. Between 150 and 280 days of pregnancy, total estrogens concentration in mare should exceed 1000 ng/mL, whereas, in the last months of pregnancy, estrogens decrease gradually to baseline concentrations [15]. Relaxin is detectable in the blood after the 80th day of pregnancy, and its pre-partum concentration ranges from 4 to 7 ng/mL [20]. A progressive rise to peak concentrations around 200 days of gestation is followed by a decline over the next 60 days [21]. Maternal concentrations of relaxin rise again in late gestation and increase further during parturition [20].

This preliminary study aimed to assess changes in thermal features of the abdomen lateral surface and concentrations of reproductive hormones in Polish native pregnant mares and compare to those found in non-pregnant mares subjected to the same conditions. The thermal pattern of the mares’ abdomens and the corresponding hormonal profile of Polish native Konik Polski mares were investigated and compared over consecutive months to find associations.

## 2. Materials and Methods 

### 2.1. Animals

The research was carried out at stud Dobrzyniewo, which is a Polish state stud farm running conservation breeding of Konik Polski horses, according to protocol approved by the II Local Ethical Committee on Animal Testing in Warsaw (Permit Number: WAW2/007/2020 from 15 January 2020) on behalf of the National Ethical Committees on Animal Testing. For the herd of 90 Konik Polski horses, 26 mares (mean age 6.28 ± 4.04 years; mean height 142.40 ± 2.12 cm), after natural mating in February and March, were selected and included into the pregnant mares group (*n* = 26). Another 14 non-lactating mares (mean age 5.47 ± 3.90 years; mean height 143.10 ± 2.09 cm), which were not mated this reproductive season, were qualified for the non-pregnant control group (*n* = 14). All mares were housed in stalls with the same management in the all-day open stable. The horses were fed twice a day with a dose of hay personalized to each horse to maintain optimal, healthy condition and had daily access to a large area grassy pasture no shorter than 12 h per day. At the beginning of research, physical examinations were conducted to ensure that the mares were free from clinical symptoms of any disease and only healthy mares were included in the study groups. The basic clinical examination—including evaluation of internal temperature, heart rate, respiratory rate, mucous membranes, capillary refill time, and lymph nodes—was carried out following international veterinary standards. Then the detailed reproductive tract examination was conducted following standard protocol [12]. Pregnancies were confirmed at 14 and 35 days post-ovulation in 26 mares in the pregnant group. In control group, pregnancy was excluded at the beginning of the research in 14 mares. Those mares were introduced into the non-pregnant group. Ultrasound examination of the reproductive tract was done with an ultrasound scanner (MyLabOne; ESAOTE, Florence, Italy) using a linear 5 MHz transducer (ESAOTE, Florence, Italy). Basing on the examination at the beginning of research, the months of gestation were defined as follows: month 1: 1–30 days; month 2: 30–60 days; month 3: 60–90 days; month 4: 90–120 days; month 5: 120–150 days; month 6: 150–180 days; month 7: 180–210 days; month 8: 210–240 days; month 9: 240–270 days; month 10: 270–300 days; and month 11: 300–330 days. Blood samples and thermal images were then collected from all mares, pregnant and non-pregnant, once a month from February until January, when the last foaling of the examined mares took place.

### 2.2. Reproductive Hormone Concentrations

All blood samples were acquired by a jugular venipuncture using a BD Vacutainer® system with dry tubes (BD Plymouth, Plymouth, UK). Blood samples were cooled to + 4 °C and transported within 5 h to the lab. Then, tubes were centrifuged (2000× *g*, 5 min), and serum free from any apparent hemolysis was aspirated for further analyses. Reproductive hormone concentrations were measured using immunoenzymatic equine species-specific commercial assays (ELK Biotechnology, Wuhan, China). Progesterone (P4; immune cross-reactivity for pregn-4-ene,3,20-dione and 5α-dihydroprogesterone) (ELK8549; ELK Biotechnology, Wuhan, China; sensitivity 0.43 ng/mL, intraassay CV < 5.8% and interassay CV < 9.2%), 17-ß estradiol (E2) (ELK8547; ELK Biotechnology, Wuhan, China; sensitivity 4.72 pg/mL, intraassay CV < 8.2% and interassay CV < 11.2%), estrone sulfate (E1S) (ELK8548; ELK Biotechnology, Wuhan, China; sensitivity 0.55 ng/mL, intraassay CV < 8.4% and interassay CV < 12.5%), and relaxin (RLN) (ELK8550; ELK Biotechnology, Wuhan, China; sensitivity 0.97 ng/mL, intraassay CV < 10.0% and interassay CV < 12.8%) commercial assays and sample dilution recommended by the manufacturer’s protocol were used. The absorbance was measured by Multiscan Reader (Labsystem, Helsinki, Finland) using a Genesis V 3.00 software. All measurements (standard and samples) were performed in duplicates, and average values were calculated. All templates were performed on one day in one incubation period to avoid influence of different humidity and temperature.

### 2.3. Thermal Imaging

The thermographic examinations were performed indoors in stable, in areas sheltered from the sunlight, in the absence of air drafts. The horses were led into the stables no earlier than two hours before imaging began. All the imaged area, the lateral surface of the mares’ abdomens, was brushed for dirt and mud removal 15 minutes before imaging. A total of 880 images were taken in a closed space, protected from wind and sun radiation, to minimize the influence of external environmental conditions [10]. The ambient temperature (Tamb) and humidity were measured and noted directly before first imaging each research day. Images were taken on the right and left side of the mare’s body using a non-contact thermographic camera (FLIR Therma CAM E60; FLIR Systems Brasil, Sorocaba, Brazil; emissivity (e) 0.99; temperature range between 10.0 and 40.0 °C). The camera was placed on a distance of approximately 2.0 m up from the imaging area and positioned on a halfway up the vertical line through the tuber coxae. The distance was standardized using a two-meter-long portable light distance indicator placed on the ground between the researcher and the mare, and the camera position was standardized using a red light collimator built-in thermal camera. All thermographic images were obtained by the same researcher (MM). All images were carried out following equine international veterinary standards [22].

The surface body temperatures were evaluated using two palettes adjusted to the same image. Firstly, the Rainbow HC palette was used to mark two measuring regions Px1 (the whole area of the lateral surface of the abdomen) and Px2 (the flank area of the lateral surface of the abdomen) (Figure 1A). Then, the Gray palette was used to visualize the lateral surface of the abdomen with marked areas at a certain temperature (Figure 1B).

The images in the Rainbow HC palette were corrected using the digital enhancement of the details function. Then, Px1 was annotated using a measurement polygon tool from the vertical line lying behind the withers to the vertical line behind the tuber coxae. Px1 region was limited at the top by the edge of the back and the bottom by the edge of the abdomen. Next, Px2 was annotated in the flank area by the measurement magic tool. Px2 region was limited by the vertical line behind the tuber coxae and by the edge of the last rib. The Px2 height was the lower 2/3 of the Px1 height. The maximal temperature in Px1 and Px2 (Tmax), the average temperature (Taver) of each Px1 and Px2, and the minimal temperature (Tmin) of each Px1 and Px2 were calculated using professional software SENSE Batch (SENSE Software, Warsaw, Poland). Tmax represented the values of the highest temperatures recorded in both Px1 and Px2, because Px1 included Px2, and their maximum temperatures are equal to one another. Taver reported the value of the mean temperature calculated for the entire Px’s area, whereas Tmin reported the values of the lowest temperatures recorded in Px1 and Px2, respectively. Obtained data were presented in the form of a data series, in which subsequent horses were represented by other realizations. 

On the images in the Gray palette, the area with the highest temperatures in the range of 3 °C (Area of Tmax) was annotated in red and yellow using professional software FLIR Tools Professional (FLIR Systems Brasil, Sorocaba, Brazil). The range was adjusted to consecutive mares based on the individual Tmax level. The Gray palette images with the Area of Tmax annotated in red were used to prepare visualization of general trends (Figure 1B), whereas images with the Area of Tmax annotated in yellow were used in procedure for counting the annotated pixels in thermograms (Figure 1C). In the counting procedure, the input images were grayscale RGB images with the Area of Tmax annotated in yellow. In order to count the pixels in annotated areas, they were segmented by thresholding the three color channels (red, green, blue), which resulted in a binary image. The pixel was assigned the value 1 if the values of its channels pr < 0.8 and pg < 0.8 and pb > 0.5; otherwise, it was assigned the value of 0. Finally, pixels with non-zero value were counted (Figure 1D). The results were presented as the ratio of the number of yellow annotated pixels to the total number of pixels expressed as a percentage. The pixels counting experiment was implemented in Python 3.6.9.

### 2.4. Statistical Analyses

Univariate marginal distributions of hormone concentrations and thermal data were tested independently for each month of study using a Shapiro–Wilk normality test. Repeated measures one-way ANOVA and the Friedman test were performed to compare the data series as paired data between months of study. The comparison of data showing normal distribution was assessed by a Repeated measures one-way ANOVA with Geisser–Greenhouse correction, followed by Tukey’s multiple comparisons test. The non-Gaussian data was evaluated by the Friedman test, followed by Dunn’s multiple comparisons test. Unpaired t-test with Welch’s correction and the Mann–Whitney test were performed to compare the data series as unpaired data between pregnant and non-pregnant groups. Data showing normal distribution was compared by an Unpaired t-test with Welch’s correction, whereas non-Gaussian data was compared by the Mann–Whitney test. Numerical data, excepting Tamb, were presented in tables as the mean ±SD. Linear regressions of the measurements across months of pregnancy and corresponding months in non-pregnant group were calculated for following pairs of data series: each hormone concentration (P4, E1S, E2, REL) and each thermal feature (Tmax, Taver in Px1, Taver in Px2, Tmin in Px1, Tmin in Px2, Area of Tmax). The regression equation and r square were presented on each plot. All the slopes were significantly non-zero (*p* < 0.0001). Then, the slopes in hormone concentration equation and in thermal features equation were compared. When differences between the slopes were not significant (*p* > 0.05), one slope for all the data was calculated, and then the intercepts in hormone concentration equation and in thermal features equation were compared. All statistical analyses were performed using GraphPad Prism6 software (GraphPad Software Inc., San Diego, CA, USA), where the significance level was established as *p* < 0.05.

## 3. Results

Thermal imaging was conducted for 11 repetitions, with ambient temperatures ranging from 1.0 °C to 24.0 °C and the humidity ranging from 50 to 90%. The descriptive statistics (mean ±SD) for selected thermal features of the lateral surface of the abdomen and reproductive hormone concentrations in the serum are presented in Table 1 and Table 2, respectively. The values of thermal features of the lateral surface of the abdomen differed significantly (*p* < 0.0001) between studied months. The lowest values occurred not only in the first two months but also in the last two months of pregnancy, with Tamb below 10.0 °C. In the pregnant group, the maximal level of Tmax and Taver Px2 were noted between the fourth and seventh month of pregnancy, whereas that of Taver Px1, Tmin Px1, and Tmin Px2 were between the sixth and seventh month of pregnancy. In the non-pregnant group, the maximal level of Tmax was noted between the third and seventh months, Taver Px1 and Taver Px2 were between the fourth and seventh months, and Tmin Px1 and Tmin Px2 were between the fifth and seventh months. The highest Tamb (above 20.0 °C) was also noted during months six and eight. Then, measured temperatures decreased gradually, similarly to the decrease in Tamb (from 21.2 °C at the eight month to 1.0 °C at the eleventh month). In the pregnant group, the lowest values of Tmax, Taver Px1, and Tmin Px1 were recorded in the tenth and eleventh month of pregnancy, and no difference with the first and second months of pregnancy was found. The values of Taver Px2 and Tmin Px2 decreased to a level higher than in the first and second month. In non-pregnant group, the lowest values of Tmax were recorded in tenth and eleventh months; Taver Px1 in the first and from the ninth to eleventh months; Taver Px2 in the first, second, third, tenth, and eleventh months; and Tmin Px1 and Tmin Px2 in the first, second, tenth and eleventh months. The Tmax was higher in the pregnant than non-pregnant group between the sixth and eleventh months. Similarly, Taver Px1 was higher between the sixth and eight months, and Taver Px2 between the sixth and eleventh months. The Tmin Px1 was higher in the pregnant than non-pregnant group only in eight month, whereas Tmin Px2 was between the eight and eleventh months (Table 1). With measured thermal features, only Area of Tmax showed no monthly changes similar to changes in Tamb. In the pregnant group, the low Area of Tmax was observed between the first and sixth month of pregnancy. Then Area of Tmax increased gradually to the maximal value achieved in the eight month and remained high until the eleventh month of pregnancy. In the non-pregnant group, no differences in the Area of Tmax were found between the studied months. The Area of Tmax was higher in the pregnant than non-pregnant group between the sixth and eleventh months (Table 1; Figure 2). 

In the pregnant group, months with the highest values of measured temperatures, both ambient and surface, corresponded with the highest E1S concentration. In the pregnant group, the E1S concentration increased gradually throughout the first four months of pregnancy with a significant increase at month five and was maintained above 100 ng/mL from the sixth to nineth month. Then, a significant decrease to the initial baseline was noted. In the non-pregnant group, no differences in E1S concentrations were found between the studied months. The E1S concentration was higher in pregnant than non-pregnant groups between the second and tenth months (Table 2). In the pregnant group, the concentration of E2 was low in the first month, then increased during the second month to the highest concentration above 100 pg/mL at the third month. The gradual decrease in E2 concentration was noted at the fourth and fifth months to the initial baseline comparable to the first month. Some E2 concentration decreases were noted at the ninth and eleventh month of gestation (Table 2). In the non-pregnant group, no differences in E2 concentrations were found between the studied months. The E2 concentration was higher in the pregnant than non-pregnant group between the second and fifth months. No evidence of association between E1S concentration (Figure 3 and Figure 4) or E2 concentration (Figure 5 and Figure 6) and thermal features of the lateral surface were observed both in pregnant and non-pregnant groups.

In the pregnant group, the mean P4 concentration was above 8.4 ng/mL throughout the pregnancy. Some fluctuations in P4 concentration were noted between the first and eighth months; however, the significant increase in P4 concentration occurred in the ninth, tenth, and eleventh months. In the non-pregnant group, no differences in P4 concentrations were found between the studied months. The P4 concentration was higher in the pregnant than non-pregnant group between the first and eleventh months. In the pregnant group, the REL concentration was not detectable in the first and second month. Then, REL concentration reached the level of 7.2–16.9 ng/mL and remained without significant differences to month five. The REL concentrations were also high in month 6 of pregnancy (and like those of months 10 and 11), when the lowest Area of Tmax was observed. In the non-pregnant group, the REL concentration was not detectable. Interestingly, the months with the highest values of Area of Tmax corresponded with the highest P4 and REL concentrations. The slopes in the linear regression equation were significantly no different for P4 concentration and Area of Tmax (*p* = 0.9245) as well as REL concentration and Area of Tmax (*p* = 0.4324). The intercepts in the linear regression equation were significantly different for P4 concentration and Area of Tmax (*p* < 0.0001) as well as REL concentration and Area of Tmax (*p* < 0.0001). One slope for P4 concentration and Area of Tmax and REL concentration and Area of Tmax was established as at the level of 1.373 and 1.342, respectively. In the pregnant group, no evidence of association between P4 concentration (Figure 7) or REL concentration (Figure 8) and measured temperatures of the lateral surface of the abdomen were observed. Similarly, in the non-pregnant group, no evidence of association between P4 concentration (Figure 9) and all thermal features of the lateral surface were observed. 

## 4. Discussion

Bowers et al. reported that the differences in superficial body temperature between pregnant and non-pregnant mares allow for the detection of pregnancy in the horse during late gestation. The authors suggested that the changes in mares’ flank temperatures may be caused by the generalized hormonal effects on regional blood flow [1]. Since the blood flow in the flank skin and subcutaneous tissue is difficult to evaluate, this preliminary study aimed to initially assess the association between reproductive hormone concentrations and thermal features of the lateral surface of the mares’ abdomens. Our results indicated that the surface body temperature, both in the whole area and the flank area of the lateral surface of the abdomen, changes similarly to Tamb; however, no correlations were measured. This observation, in both pregnant and non-pregnant groups, is in line with previous thermographic studies, where ambient temperature was reported to significantly affect the superficial body temperature [10,11,23]. Moreover, in the study, the background temperature was positively correlated with the flank and withers temperature, regardless of group (pregnant, non-pregnant, foaled) and was positively correlated with ambient temperature [1]. Based on the thermographic imaging of flank area, when pregnant and non-pregnant mares were compared and imaged under stable environmental conditions, it was possible to distinguish between pregnant and non-pregnant mares in the tenth and eleventh months of pregnancy [1]. Bowers et al. observed differences in the flank temperature between pregnant and non-pregnant mares in the tenth and elevent months of pregnancy [1]. In this study, similar differences were noted already from the sixth month (Area of Tmax, Tmax, Taver). In the tenth and eleventh months of pregnancy, Tmin Px2 difference was also observed between groups. Since Bowers et al. did not measure temperature in the earlier months of pregnancy and only assessed average flank temperatures [1], the results obtained broaden the current knowledge on the lateral thermal pattern in pregnant mares. In the present study, we observed Area of Tmax increase gradually as pregnancy progressed, and, unlike measured temperatures, it did not decrease as Tamb decreased. For this reason, Area of Tmax may be better than Tmax, Taver, or Tmin for thermographic evaluation of pregnant mares. However, certain distinguishing between pregnant and non-pregnant mares using an approach based on Area of Tmax calculation requires further research.

In the Bowers et al. study, thermographic imaging was confirmed to detect pregnancy in the horse during late gestation [1]. In this research, infrared thermography has not been studied as a pregnancy diagnosis tool. This preliminary study aimed to introduce the use of different thermographic palettes (rainbow HC; gray palette) and the maximum temperature pixel count (Area of Tmax) into the pregnant mare thermographic imaging. Maximum temperature pixel counts seem to be an interesting approach not only because of the lower dependence on ambient temperature but also because of the possible association between Area of Tmax and the concentration of pregnancy hormones in the blood. In the pregnant group, we demonstrated the association between Area of Tmax as well as P4 and REL concentration across months of pregnancy and no similar evidence between the superficial body temperatures and reproductive hormone concentrations. Additionally, in the non-pregnant group, no similar association was noted. Obtained results of the linear regression indicate the potential possibility of predicting P4 and REL concentrations based on the size of the lateral surface of the abdominal area at high temperature. In the study presented here, the day of breeding was known, and the mares were monitored over time. In a case of wild horses, the ability to determine pregnancy is limited by lack of knowledge about fact and time of mating [24]; seasonal changes in the hair coat lengths [25,26], which may affect the thermal properties of the skin and hair coat and, consequently, thermographic images [9,27]; and finally, the environmental conditions during thermographic imaging [11,12,23]. Therefore, further research involving measuring the hair coat lengths [9], comparison of thermal features of brushed and non-brushed lateral surface of abdomen [5], and comparison of indoor and outdoor thermographic imaging [28] are required to determine whether the measurement of Area Tmax of mares’ flank may be further investigated as a potential pregnancy diagnosis tool. 

The main limitation of the presented study is the monthly examinations and blood sampling due to COVID-19 restrictions. Therefore, this study is a preliminary study to monitor the changes and associations of the thermal characteristics of the lateral surface of the abdomen and the concentrations of reproductive hormones in pregnant Polish native mares. The recommended frequency of blood sampling varies from three times a day [18] to once every 2-3 days [19], whereas in this study, samples were taken once a month. Since E1S, E2, and partially P4/DHP have a pulsatile release, the maximal measured E1S concentration did not exceed 1000 ng/mL recommended for total estrogens between 150 and 280 days of pregnancy [15], whereas the maximal E2 concentration occurred between 60 and 90 days of pregnancy [29,30]. However, no evidence of pulsatile releases of E1S and E2 during the last week of pregnancy was noted [31], due to sparse sampling. Progesterone concentrations, independently from source changing from corpora lutea (P4) to the fetoplacental unit (DHP), should be above 4 ng/mL throughout pregnancy [30,31,32]. If the entire pregnancy is screened, the monthly blood sampling would allow us to show the gradual P4/DHP increase in the second and third trimester up to the concentration of 12 ng/mL [19]; however, the progesterone peak of P4/DHP at the last week before parturition [18] was omitted. Similarly, a progressive REL concentration rises to peak at around 200 days of pregnancy [21], with REL concentration higher than 7 ng/mL [20], which was possible to observe under monthly blood sampling. Both P4/DHP and REL concentrations were predicted as higher than Area of Tmax 1.373 and 1.342 times, respectively. Bearing in mind the discussed limitations, further research with at least daily blood sampling and corresponding thermographic imaging is required to consider Area of Tmax calculation as a potential pregnancy diagnosis tool. This step is essential to better understanding the relation between thermographic imaging and concentrations of reproductive hormones in pregnant mares.

## 5. Conclusions

In the present study, individual thermal features of the abdomen lateral surface differed between pregnant and non-pregnant mares in other periods. Differences in maximal and average temperature as well as the area with the highest temperatures were observed from the sixth month of pregnancy, whereas differences in minimal temperature were observed from the eighth month of pregnancy. Measures of superficial body temperatures have been found to change monthly, similarly to ambient temperatures, with no evidence of coincidence with changes in reproductive hormone concentrations. The area with the highest temperatures of the lateral surface of the abdomen was less susceptible to the influence of ambient temperature and associated across months of pregnancy with serum concentrations of progesterone and relaxin. Further research is needed to better understand the relation between thermographic imaging, especially an approach based on Area of Tmax calculation, and concentrations of reproductive hormones in pregnant mares.

## Figures and Tables

**Figure 1 animals-11-01517-f001:**
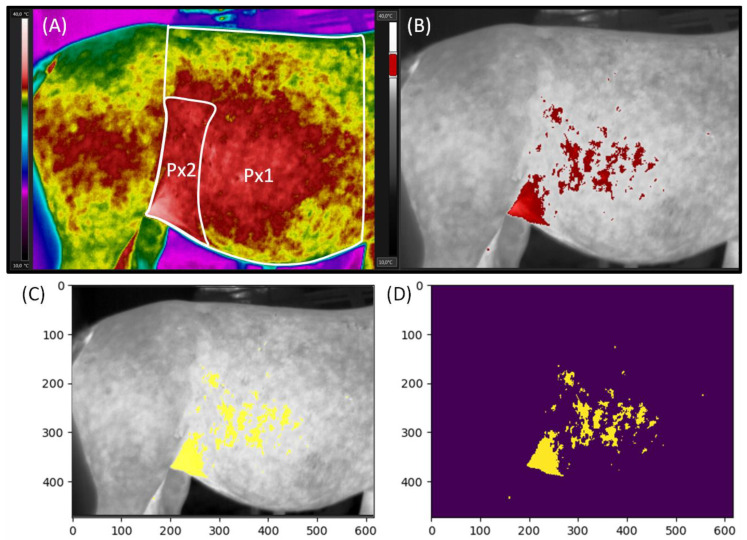
Thermal images of the lateral surface of the abdomen of the pregnant mare. Rainbow HC palette with marked Px1 and Px2 (**A**). Gray palette with red annotated areas at a certain temperature (**B**). Gray palette with yellow annotated areas at a certain temperature in the pixel counting procedure (**C**). The counted pixels with non-zero value (**D**). Abbreviations: Px1, the whole area of the lateral surface of the abdomen; Px2, the flank area of the lateral surface of the abdomen.

**Figure 2 animals-11-01517-f002:**
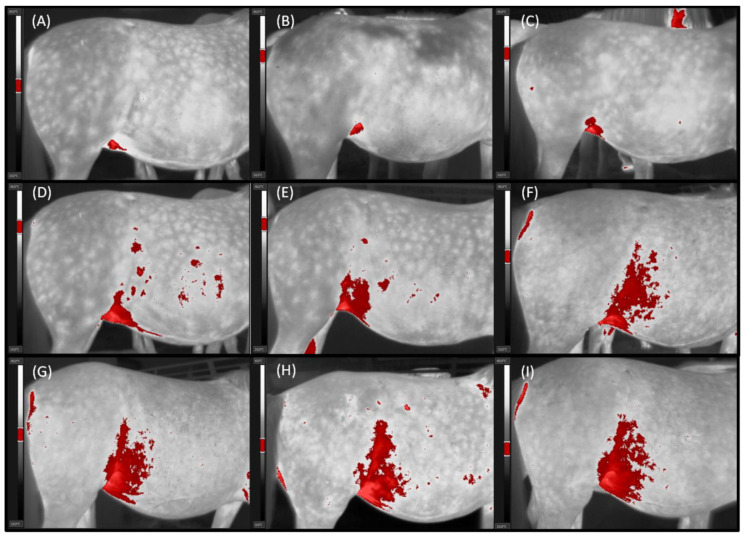
Thermal images in gray palette of the mare. The area with the highest temperatures in the range of 3 °C (Area of Tmax) marked in red in the following months of pregnancy: third (**A**), fourth (**B**), fifth (**C**), sixth (**D**), seventh (**E**), eighth (**F**), ninth (**G**), tenth (**H**), and eleventh (**I**).

**Figure 3 animals-11-01517-f003:**
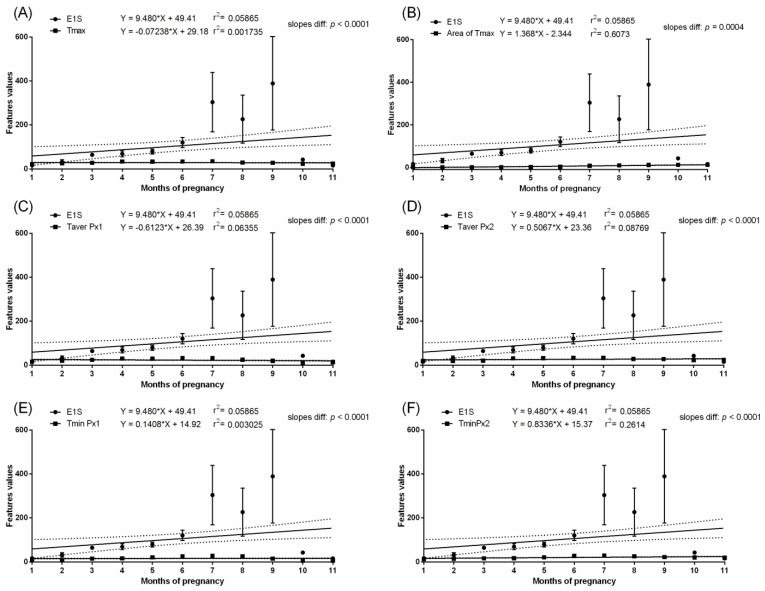
Linear regressions of estrone sulfate concentration in the serum and selected thermal features of the lateral surface of the abdomen in pregnant mares: the maximal temperature in Px1 and Px2 (**A**); the area with the highest temperatures in the range of 3 °C (**B**); the average temperature in Px1 (**C**); the average temperature in Px2 (**D**); the minimal temperature in Px1 (**E**); the minimal temperature in Px2 (**F**).

**Figure 4 animals-11-01517-f004:**
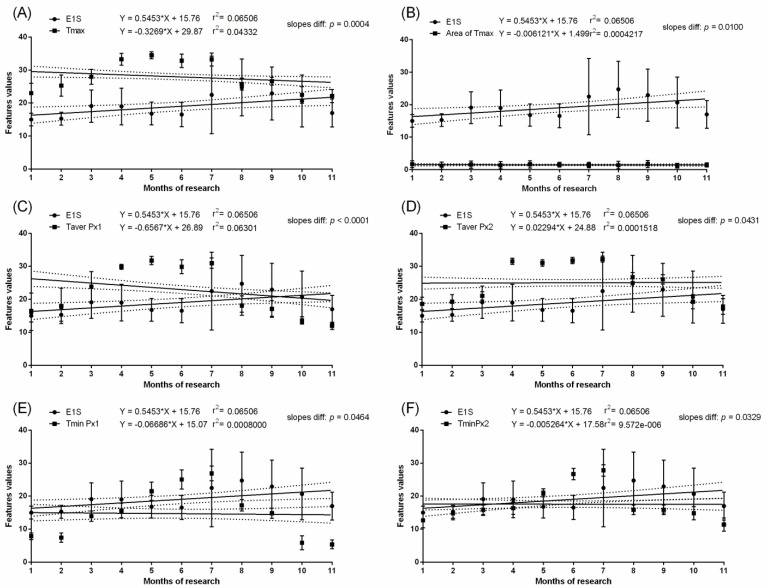
Linear regressions of estrone sulfate concentration in the serum and selected thermal features of the lateral surface of the abdomen in non-pregnant mares: the maximal temperature in Px1 and Px2 (**A**); the area with the highest temperatures in the range of 3 °C (**B**); the average temperature in Px1 (**C**); the average temperature in Px2 (**D**); the minimal temperature in Px1 (**E**); the minimal temperature in Px2 (**F**).

**Figure 5 animals-11-01517-f005:**
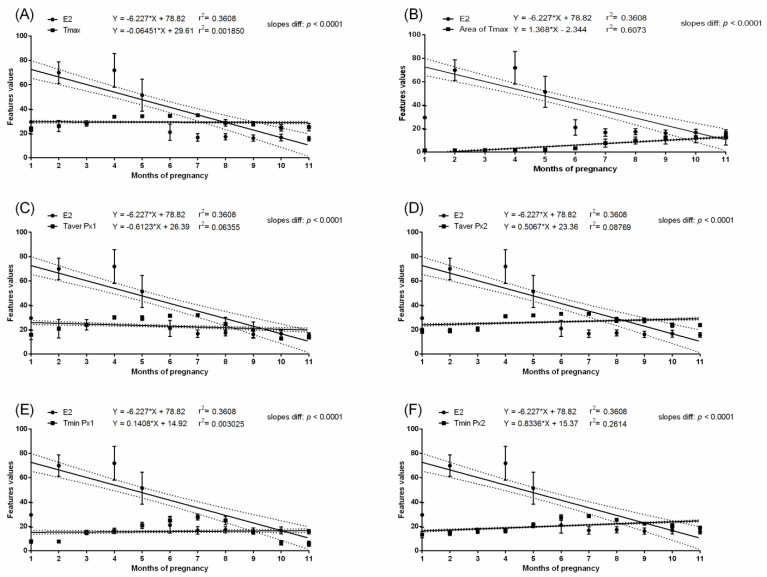
Linear regressions of 17-ß estradiol concentration in the serum and selected thermal features of the lateral surface of the abdomen in pregnant mares: the maximal temperature in Px1 and Px2 (**A**); the area with the highest temperatures in the range of 3 °C (**B**); the average temperature in Px1 (**C**); the average temperature in Px2 (**D**); the minimal temperature in Px1 (**E**); the minimal temperature in Px2 (**F**).

**Figure 6 animals-11-01517-f006:**
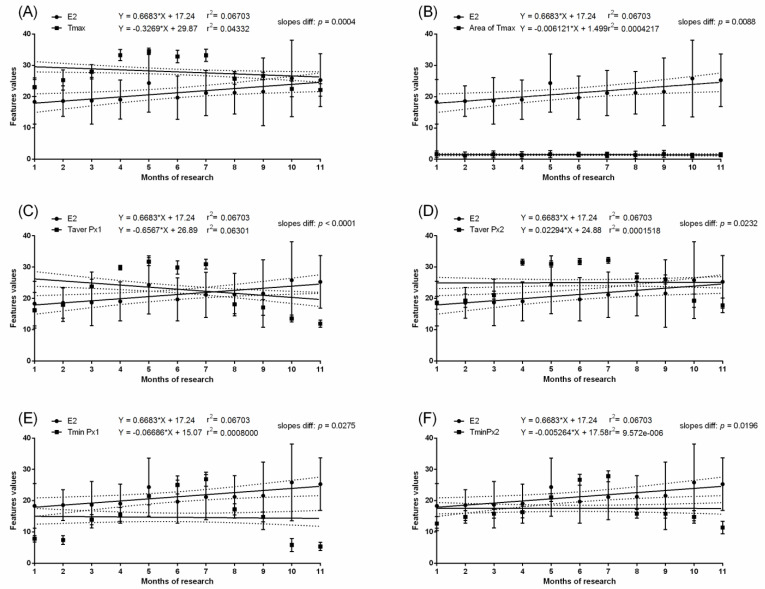
Linear regressions of 17-ß estradiol concentration in the serum and selected thermal features of the lateral surface of the abdomen in non-pregnant mares: the maximal temperature in Px1 and Px2 (**A**); the area with the highest temperatures in the range of 3 °C (**B**); the average temperature in Px1 (**C**); the average temperature in Px2 (**D**); the minimal temperature in Px1 (**E**); the minimal temperature in Px2 (**F**).

**Figure 7 animals-11-01517-f007:**
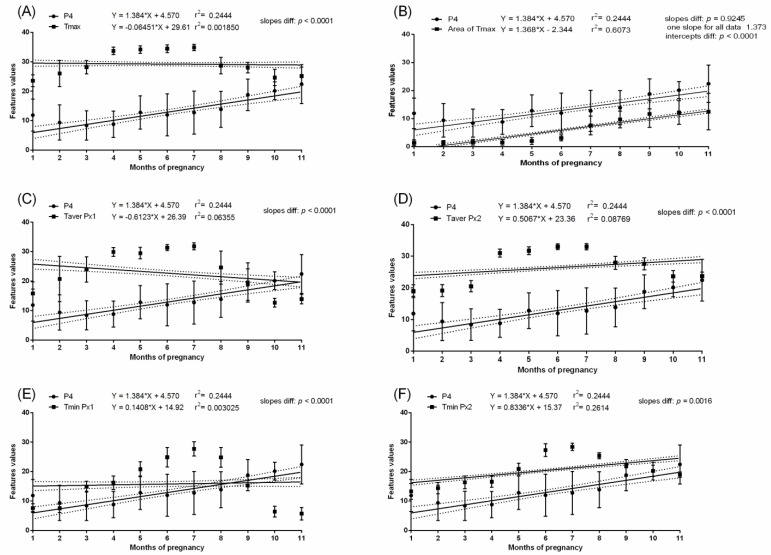
Linear regressions of progesterone concentration in the serum and selected thermal features of the lateral surface of the abdomen in pregnant mares: the maximal temperature in Px1 and Px2 (**A**); the area with the highest temperatures in the range of 3 °C (**B**); the average temperature in Px1 (**C**); the average temperature in Px2 (**D**); the minimal temperature in Px1 (**E**); the minimal temperature in Px2 (**F**).

**Figure 8 animals-11-01517-f008:**
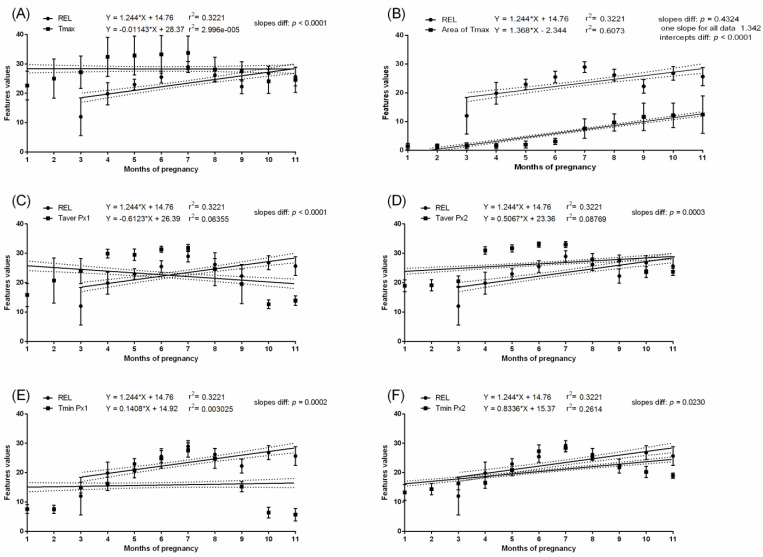
Linear regressions of relaxin concentration in the serum and selected thermal features of the lateral surface of the abdomen in pregnant mares: the maximal temperature in Px1 and Px2 (**A**); the area with the highest temperatures in the range of 3 °C (**B**); the average temperature in Px1 (**C**); the average temperature in Px2 (**D**); the minimal temperature in Px1 (**E**); the minimal temperature in Px2 (**F**).

**Figure 9 animals-11-01517-f009:**
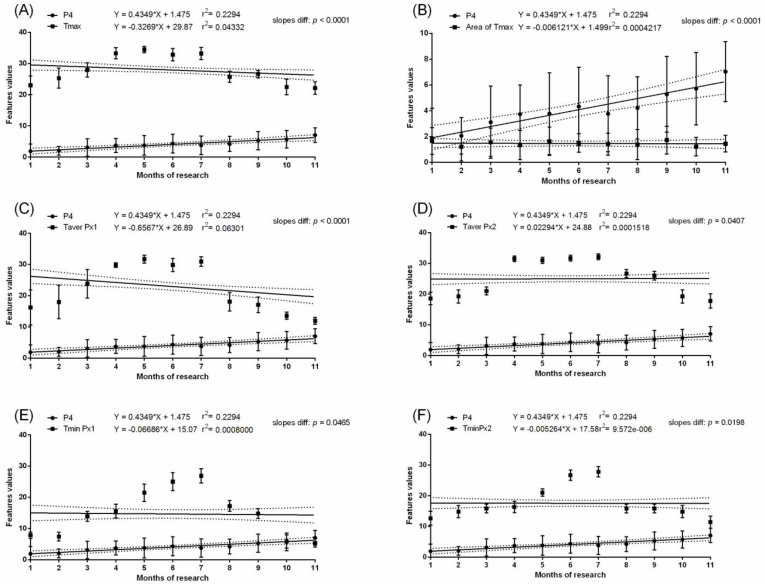
Linear regressions of progesterone concentration in the serum and selected thermal features of the lateral surface of the abdomen in non-pregnant mares: the maximal temperature in Px1 and Px2 (**A**); the area with the highest temperatures in the range of 3 °C (**B**); the average temperature in Px1 (**C**); the average temperature in Px2 (**D**); the minimal temperature in Px1 (**E**); the minimal temperature in Px2 (**F**).

**Table 1 animals-11-01517-t001:** Values of selected thermal features: maximal temperature (°C), average temperature (°C), minimal temperature (°C), and area with the highest temperatures in the range of 3 °C (%) of the lateral surface of abdomen of pregnant (P; *n* = 26) and non-pregnant (NP; *n* = 14) mares in consecutive months of pregnancy.

Mo	Group	1	2	3	4	5	6	7	8	9	10	11
Tmax	P	23.5 ± 2.0 ^a^	26.0 ± 4.5 ^ac^	28.2 ± 2.3 ^bc^	33.7 ± 1.3 ^b^	34.2 ± 1.3 ^b^	34.5 ± 1.3 ^b^	35.9 ± 1.1 ^b^	28.7 ± 2.8 ^c^	28.0 ± 1.8 ^c^	24.7 ± 2.8 ^ac^	25.1 ± 3.1 ^ac^
	NP	23.0 ± 2.9 ^x^	25.3 ± 3.2 ^x^	27.9 ± 2.3 ^y^	33.3 ± 1.8 ^y^	34.5 ± 1.0 ^y^	32.8 ± 2.0 ^y,^*	33.2 ± 1.9 ^y,^*	25.7 ± 1.8 ^x^	26.6 ± 1.2 ^x^	22.5 ± 2.5 ^xz^	22.1 ± 2.0 ^z^
Taver Px1	P	15.8 ± 4.0 ^a^	20.8 ± 7.7 ^ab^	24.0 ± 4.3 ^abc^	29.9 ± 1.5 ^b^	29.5 ± 2.0 ^b^	31.4 ± 1.0 ^bd^	31.8 ± 1.2 ^bd^	24.6 ± 5.6 ^b^	19.6 ± 6.7 ^ac^	12.7 ± 1.5 ^a^	13.9 ± 1.6 ^a^
	NP	16.2 ± 5.7 ^x^	18.0 ± 5.4 ^xy^	23.8 ± 4.6 ^xy^	29.8 ± 0.7 ^yz^	31.7 ± 1.2 ^z^	29.9 ± 2.1 ^z,^*	30.9 ± 1.5 ^z,^*	18.1 ± 5.6 ^xz,^*	17.1 ± 2.5 ^x^	13.6 ± 1.1 ^x^	11.9 ± 1.1 ^x^
Taver Px2	P	18.9 ± 2.0 ^a^	19.1 ± 1.9 ^ab^	20.5 ± 1.8 ^ab^	31.0 ± 1.3 ^cd^	31.7 ± 1.2 ^cd^	33.0 ± 0.9 ^c^	32.9 ± 1.0 ^c^	27.9 ± 2.1 ^de^	27.4 ± 2.0 ^de^	23.6 ± 1.8 ^be^	23.7 ± 1.2 ^be^
	NP	18.6 ± 2.0 ^x^	19.3 ± 2.2 ^x^	21.0 ± 1.3 ^x^	31.5 ± 0.9 ^yz^	31.0 ± 1.0 ^yz^	31.8 ± 0.9 ^yz,^*	32.2 ± 0.9 ^yz,^*	26.7 ± 1.3 ^xz^	26.0 ± 1.4 ^xz^	19.3 ± 2.2 ^x^	17.8 ± 2.4 ^x^
Tmin Px1	P	7.6 ± 1.5 ^a^	7.6 ± 1.4 ^a^	14.9 ± 1.8 ^b^	16.2 ± 2.3 ^b^	20.8 ± 2.6 ^bcd^	24.7 ± 3.5 ^c^	27.7 ± 2.4 ^c^	24.6 ± 3.3 ^c^	15.2 ± 1.8 ^bd^	6.4 ± 1.8 ^a^	5.7 ± 2.1 ^a^
	NP	7.8 ± 1.0 ^x^	7.4 ± 1.4 ^x^	13.9 ± 1.6 ^xy^	15.6 ± 2.2 ^xz^	21.4 ± 2.7 ^yz^	25.0 ± 2.8 ^yz^	26.9 ± 2.3 ^z^	15.7 ± 1.4 ^xz,^*	14.8 ± 1.5 ^xz^	5.8 ± 2.1 ^x^	5.4 ± 1.3 ^x^
Tmin Px2	P	13.3 ± 2.7 ^a^	14.4 ± 1.9 ^a^	16.3 ± 2.2 ^ab^	16.5 ± 1.8 ^ab^	20.8 ± 1.9 ^c^	27.3 ± 2.2 ^d^	28.4 ± 1.2 ^d^	25.4 ± 1.0 ^cd^	22.1 ± 0.8 ^c^	20.1 ± 1.9 ^bc^	19.0 ± 1.0 ^bc^
	NP	12.6 ± 2.3 ^x^	14.8 ± 2.0 ^x^	15.8 ± 1.4 ^xy^	16.3 ± 1.9 ^xy^	21.0 ± 1.2 ^xyz^	26.7 ± 1.7 ^z^	27.8 ± 1.7 ^z^	15.8 ± 1.2 ^xv,^*	15.8 ± 1.4 ^xv,^*	14.8 ± 2.0 ^x^	11.4 ± 2.0 ^x^
Area of Tmax	P	1.4 ± 1.0 ^a^	1.4 ± 0.8 ^a^	1.6 ± 1.0 ^a^	1.5 ± 1.0 ^a^	2.0 ± 1.2 ^a^	3.1 ± 1.1 ^ac^	7.6 ± 3.3 ^bc^	9.7 ± 3.0 ^b^	11.7 ± 4.7 ^b^	12.2 ± 4.3 ^b^	12.5 ± 6.5 ^b^
	NP	1.7 ± 1.1 ^x^	1.2 ± 1.1 ^x^	1.5 ± 1.1 ^x^	1.3 ± 1.2 ^x^	1.6 ± 1.1 ^x^	1.5 ± 0.7 ^x,^*	1.4 ± 0.8 ^x,^*	1.4 ± 1.2 ^x,^*	1.7 ± 1.1 ^x,^*	1.2 ± 0.7 ^x,^*	1.5 ± 0.6 ^x,^*
Tamb	-	6.1	6.9	14.3	13.0	18.2	20.4	24.0	21.2	12.1	4.1	1.0

Note: Different superscripts within the same row are statistically different (*p* < 0.05). Differences between pregnant and non-pregnant groups are marked with an asterisk in the NP row (* *p* < 0.05). Data are shown as mean ± SD. Abbreviations: Mo, months of pregnancy and corresponding months in the non-pregnant group; Px1, the whole area of lateral surface of abdomen; Px2, the flank area of lateral surface of abdomen; Tmax, maximal temperature in Px1 and Px2; Taver Px1, average temperature in Px1; Taver Px2, average temperature in Px2; Tmin Px1, minimal temperature in Px1; Tmin Px2, minimal temperature in Px2; Tamb, the ambient temperature; Area of Tmax, the area with the highest temperatures in the range of 3 °C.

**Table 2 animals-11-01517-t002:** Concentrations of selected reproductive hormones: progesterone (ng/mL), estrone sulfate (ng/mL), 17-ß estradiol (pg/mL), and relaxin (pg/mL) in serum of pregnant (P; *n* = 26) and non-pregnant (NP; *n* = 14) mares in consecutive months of pregnancy.

Mo	Group	1	2	3	4	5	6	7	8	9	10	11
P4 (ng/mL)	P	11.9 ± 5.4 ^abc^	9.4 ± 6.0 ^ac^	8.4 ± 5.0 ^a^	8.8 ± 4.4 ^a^	12.8 ± 5.6 ^abc^	12.0 ± 7.2 ^ac^	12.7 ± 13.9 ^abc^	13.9 ± 6.1 ^abc^	18.8 ± 5.3 ^bd^	20.2 ± 3.0 ^cd^	22.4 ± 6.6 ^cd^
	NP	1.8 ± 2.3 ^x,^*	2.0 ± 1.4 ^x,^*	3.1 ± 2.8 ^x,^*	3.7 ± 2.5 ^x,^*	3.8 ± 3.1 ^x,^*	4.3 ± 3.0 ^x,^*	3.8 ± 2.9 ^x,^*	4.2 ± 2.1 ^x,^*	5.3 ± 2.9 ^x,^*	5.7 ± 2.8 ^x,^*	7.0 ± 5.1 ^x,^*
E1S (ng/mL)	P	16.7 ± 3.4 ^a^	31.9 ± 10.1 ^ac^	64.2 ± 7.6 ^ab^	69.0 ± 13.6 ^ab^	79.2 ± 11.3 ^bc^	120.5 ± 23.0 ^b^	304.0 ± 135.0 ^b^	226.5 ± 109.6 ^b^	389.1 ± 212.2 ^b^	42.6 ± 7.7 ^ac^	15.7 ± 3.0 ^a^
	NP	15.0 ± 1.9 ^x^	15.3 ± 1.9 ^x,^*	19.1 ± 4.9 ^x,^*	18.9 ± 5.6 ^x,^*	16.8 ± 3.4 ^x,^*	16.5 ± 3.7 ^x,^*	22.5 ± 11.7 ^x,^*	24.7 ± 8.6 ^x,^*	22.9 ± 8.0 ^x,^*	20.7 ± 7.8 ^x,^*	16.9 ± 4.2 ^x^
E2 (pg/mL)	P	29.5 ± 10.1 ^a^	69.9 ± 8.8 ^b^	102.3 ± 16.4 ^c^	71.8 ± 13.8 ^d^	51.3 ± 13.2 ^e^	21.0 ± 6.6 ^af^	16.7 ± 3.1 ^af^	17.2 ± 2.5 ^af^	16.1 ± 2.5 ^f^	16.6 ± 2.8 ^af^	15.6 ± 1.7 ^f^
	NP	18.3 ± 7.1 ^x^	18.6 ± 4.9 ^x,^*	18.7 ± 7.4 ^x,^*	19.1 ± 6.3 ^x,^*	24.3 ± 9.3 ^x,^*	19.7 ± 6.9 ^x^	21.1 ± 7.2 ^x^	21.2 ± 6.9 ^x^	21.6± 10.8 ^x^	25.8 ± 12.3 ^x^	25.3 ± 8.4 ^x^
REL (ng/mL)	P	ND	ND	7.2 ± 2.4 ^a^	7.9 ± 1.7 ^a^	16.9 ± 1.8 ^ab^	20.5 ± 2.0 ^bc^	21.1 ± 1.9 ^c^	20.2 ± 2.0 ^bc^	15.3 ± 2.4 ^ab^	20.8 ± 2.4 ^bc^	21.7 ± 3.2 ^bc^
	NP	ND	ND	ND	ND	ND	ND	ND	ND	ND	ND	ND

Note: Different superscripts within the same row are statistically different (*p* < 0.05). Differences between pregnant and non-pregnant groups are marked with an asterisk in the NP row (* *p* < 0.05). Data are shown as mean ±SD. Abbreviations: Mo, months of pregnancy and corresponding months in the non-pregnant group; P4, progesterone; E1S, estrone sulfate; E2, 17-ß estradiol; REL, relaxin; ND, not detectable.

## Data Availability

The data presented in this study are available on request from the corresponding author. The data are not publicly available due to the privacy policy of Polish National Stud Farm Dobrzyniewo Sp. z o.o. (Falmierowo, Poland).

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
