# Peer review of "Association between the Area of the Highest Flank Temperature and Concentrations of Reproductive Hormones during Pregnancy in Polish Konik Horses—A Preliminary Study"

_animals, 2021, doi:10.3390/ani11061517_

Round 1

Reviewer 1 Report

This study aimed to examine the association between reproductive hormone concentrations and thermal features of the lateral surface of the abdomen in pregnant Konik Polski mares. Both endocrine profile (progesterone-P4, estrone sulfate- E1S, 17-ß estradiol- E2), and relaxin- REL) and the thermal pattern of mares' abdomen were evaluated only once monthly. The superficial body temperatures changed monthly, similarly to ambient temperatures, and were not linked to hormonal changes. The area with the highest temperatures in the range of 3ºC (Area of Tmax) is proposed as a new approach to thermographic imaging analysis. By using this new approach, an association between the Area of Tmax and concentrations of P4 (the slop = 1.373; p = .9245) and REL (the slop = 1.342; p = .4324) was observed. The authors suggested that this new approach based on Area of Tmax calculation seems to be a promising method of pregnancy evaluation.

Although the manuscript focuses on a potentially interesting topic two major concerns related to the experimental design, do not allow reliable conclusions to be drawn: (i) the lack of a control group of non-pregnant mares; and (ii) the inadequate sampling frequency for endocrine assessment.

Additional data of non-pregnant and pregnant mares evaluated on the same day and in the same environment conditions should be included. More frequent blood sample collection for endocrine assessment and thermographic imaging evaluations must be included. It is essential to confirm with non-pregnant mares if the suggested new approach (Area of Tmax) is indeed a correct one for thermographic imaging analysis and if in this new approach, the defined range of 3ºC is the most appropriate and accurate one.

As an alternative, the manuscript should be rewritten and resubmitted as a preliminary follow-up study of changes in thermal features of the abdomen lateral surface and concentrations of reproductive hormones in Polish Native pregnant mares, but never as a putative method of assessing pregnancy in mares.

Author Response

Dear Reviewer,

We are grateful for the opportunity to re-submit the manuscript for consideration in the Animals. The providential critiques made are very welcome and certainly improved the quality of the manuscript. We have addressed all considerations or provided an explanation in a few replies. If there is any remaining concern, we are happy to revisit any point that the review considered necessary.

Best regards

Comment R.1.1

Although the manuscript focuses on a potentially interesting topic two major concerns related to the experimental design, do not allow reliable conclusions to be drawn: (i) the lack of a control group of non-pregnant mares; and (ii) the inadequate sampling frequency for endocrine assessment. Additional data of non-pregnant and pregnant mares evaluated on the same day and in the same environment conditions should be included. More frequent blood sample collection for endocrine assessment and thermographic imaging evaluations must be included. It is essential to confirm with non-pregnant mares if the suggested new approach (Area of Tmax) is indeed a correct one for thermographic imaging analysis and if in this new approach, the defined range of 3ºC is the most appropriate and accurate one.

Answer to R.1.1

Thank you very much for these valuable comments. The non-pregnant control group has been added and compared with the pregnant group in each of the examined months. Unfortunately, we cannot supplement the data with a higher blood sampling frequency. Due to Covid-19, the possibility of coming to the National Konik Polski horses stud, with a total of 90 horses of the same primitive native breed kept in the same conditions, was very strictly limited. Therefore, we would like to publish these results as a preliminary one based on our experience to repeat the study as soon as the pandemic restrictions were over. We also emphasized in the discussion that the limited blood sampling rate is the greatest limitation of this work and should therefore be considered as a preliminary follow-up study.

Comment R.1.2

As an alternative, the manuscript should be rewritten and resubmitted as a preliminary follow-up study of changes in thermal features of the abdomen lateral surface and concentrations of reproductive hormones in Polish Native pregnant mares, but never as a putative method of assessing pregnancy in mares.

Answer to R.1.2

Thank you very much for this recommendation. Regardless of the test results add non-pregnant mares, the manuscript has been rewritten as a preliminary follow-up study following your suggestion. We hope that the implementation of the solution you suggest will significantly improve the quality of our manuscript.

Reviewer 2 Report

Main comments

This article evaluates the differences in flank temperature in mares, before and during gestation. The authors based their research on the fact that hormonal changes associated with pregnancy will induce changes in regional blood flow (flank area), which will be evidenced by thermography. To analyze this association, the authors serially determine the concentrations of various hormones, such as P4, E2, E1S, and relaxin.

The idea of the article is interesting, since wild equids will not allow an ultrasound evaluation, as they cannot be physically contained. Thermography allows remote evaluation, without physical contact with the animal. On the other hand, the use of different thermographic palettes (rainbow HC; gray palette) is also interesting and as shown in Figure 1, the results warrant future research in this field. The maximum temperature pixel count is also an interesting approximation. For all these reasons, in my opinion, the article should be published as a preliminary approach to the use of thermography as an aid for pregnancy monitoring.

The main problem of this article is the absence of a control group of mares, which the authors could have monitored throughout their study period, subjected to the same conditions and not pregnant. The authors should validate their results by comparing with a control group. However, in defense of the results of this research, it should be noted that, although the environmental temperature was reduced, the maximum temperature and the area of maximum temperature on the flanks increased throughout gestation, which indirectly excludes the effect of environmental conditions. However, I encourage the authors to evaluate a group of non-pregnant mares during the same time period.

Minor comments:

Title: Do the authors think that there would be differences if they were horses of other races or other geographical locations? Otherwise, the inclusion of the breed of horses in the title is not needed. In addition, and considering the obtained results, it would be proper to clarity that this is a preliminary study.

Material and methods

The authors should clarify/justify why they did not have a control group, which is the main limitation of the study.

In general, the authors should further describe the standardization performed for thermographic measurements.

Thermal imaging. Thermography was performed after 15 min of grooming of the mares. Many studies recommend performing thermography after 30 min of acclimatization. Do the authors have any justification for selecting a time of 15 minutes?

How do the authors standardize the same position of the camera in relation to the animal? Small positional differences lead to significant changes in temperature, particularly if the horses cannot be manipulated.

Did you control changes in hair length? It is true that the monitored area does not usually have very long hairs, but it would be interesting if the authors included a comment on this topic.

Comment. According to figure 1, the Px1 zone seems solidly defined for its measurement. However, the Px2 area does not seem to have such a standardized determination, since the anatomical structures indicated by the authors in the description are not exactly appreciated in the thermographic image. How did the authors solve this? Do you consider that there could have been less standardization in the measurement of the Px2 area, particularly in terms of the mean temperature? -

Author Response

Dear Reviewer,

Thank you very much for your comments and the substantial amount of time invested in looking over the manuscript. We are truly grateful for your opinion that the idea of the article is interesting, the use of different thermographic palettes (rainbow HC; gray palette) is also interesting, and the maximum temperature pixel count is also an interesting approximation. It is very important to us that you appreciate all our important and valuable solutions. We have addressed all considerations or explained a few replies. If there is any remaining concern, we are happy to revisit any point that the review considered necessary.

Best regards

Comment R.2.1

The main problem of this article is the absence of a control group of mares, which the authors could have monitored throughout their study period, subjected to the same conditions and not pregnant. The authors should validate their results by comparing with a control group. However, in defense of the results of this research, it should be noted that, although the environmental temperature was reduced, the maximum temperature and the area of maximum temperature on the flanks increased throughout gestation, which indirectly excludes the effect of environmental conditions. However, I encourage the authors to evaluate a group of non-pregnant mares during the same time period.

Answer to R.2.1

Thank you very much for these valuable comments. The non-pregnant control group has been added and compared with the pregnant group in each of the examined months. We were hoping to present the results of non-pregnant mares and the ROC analysis with ultrasound in the next publication, however, we see that it is necessary now. We hope that thus the article has not become too long and has not lost its clarity and readability.

Comment R.2.2

Title: Do the authors think that there would be differences if they were horses of other races or other geographical locations? Otherwise, the inclusion of the breed of horses in the title is not needed. In addition, and considering the obtained results, it would be proper to clarity that this is a preliminary study.

Answer to R.2.2

Thank you very much for this recommendation. Now we can not answer the question would the results be different if they were horses of other races or other geographical locations. We chose Polish Native Horses - Konik Polski breed due to their close relation to extinct wild Tarpan horses. We agree with your opinion and re-written the title.

Comment R.2.3

The authors should clarify/justify why they did not have a control group, which is the main limitation of the study.

Answer to R.2.3

Again thank you for these comments. The non-pregnant control group has been added.

Comment R.2.4

In general, the authors should further describe the standardization performed for thermographic measurements.

Answer to R.2.4

Thank you for this advice. The standardization performed for thermographic measurements has been detailed as follows below.

Comment R.2.5

Thermal imaging. Thermography was performed after 15 min of grooming of the mares. Many studies recommend performing thermography after 30 min of acclimatization. Do the authors have any justification for selecting a time of 15 minutes?

Answer to R.2.5

Thank you very much for pointing this out. The acclimatization time it concerns primarily situation when horse is transported from an cold or hot environment to the area with optimal imaging temperature. The minimum recommended acclimatization time for the horse before imaging is 20 minutes [Purohit, 2009; Soroko and Howell, 2018]. Also Bowers et al. (2009) used the acclimatization time (30 minutes) as the time between leading the horse into the room and being imaged. To avoid confusion, we have added the information that the thermographic examinations were performed indoors in stable, in areas shel-tered from the sunlight, in the absence of air drafts. The horses were lead into the stables no earlier than two hours before imaging began. The dirt and mud was removed following Soroko et al. (2017) protocol.

            Soroko, M.; Howell, K.; Dudek, K. The effect of ambient temperature on infrared thermographic images of joints in the distal forelimbs of healthy racehorses. J. Therm. Biol. 2017, 66, 63-67. https://doi.org/10.1016/j.jtherbio.2017.03.018 ("Dirt and mud present in the imaging field of view was brushed away, and 15 min were allowed to pass before scanning to ensure the transient heat generated by brushing had subsided before obtaining measurements).

            We chose the shorter protocol due to the need to imaging 40 mares in the shortest possible time under the same environmental conditions. The use of a 30-minute protocol would extend the time between thermographic imaging of the first and last mare twice, which could affect the value of the obtained results.

Purohit RC. Standards for thermal imaging in veterinary medicine. Proceedings of the XIth European Congress of Thermology; 2009, Mannheim, Germany: Thermol Int; 2009;19. p. 99.

Soroko, M.; Howell, K. Infrared thermography: current applications in equine medicine. J. Equine Vet. Sci. 2018, 60, 90-96. https://doi.org/10.1016/j.jevs.2016.11.002

Bowers, S.; Gandy, S.; Anderson, B.; Ryan, P.; Willard, S. Assessment of pregnancy in the late-gestation mare using digital infrared thermography. Theriogenology 2009, 72(3), 372-377. https://doi.org/10.1016/j.theriogenology.2009.03.005

Comment R.2.6

How do the authors standardize the same position of the camera in relation to the animal? Small positional differences lead to significant changes in temperature, particularly if the horses cannot be manipulated.

Answer to R.2.6

Thank you very much for pointing this out. We agree with your opinion that small positional differences lead to significant changes in temperature. The distance between horse and camera was standardized using a two-meter long, portable light distance indicator placed on the ground between the imager and the mare. The camera position was standardized using a red light collimator built-in thermal camera. This information has been added in the materials and methods section.

Comment R.2.7

Did you control changes in hair length? It is true that the monitored area does not usually have very long hairs, but it would be interesting if the authors included a comment on this topic.

Answer to R.2.7

Yes, the hair length was measured during whole research. The hair coat samples were taken from the mid-neck approximately 5 cm below the base of the mane following protocol described by Osthaus et al. (2018). The hair coat samples including the roots were collected into the individual tubes. The individual hair length was determined from a random sample of ten pulled strands.

            However, the were hoping to present the results of the hair length and the lateral thermal pattern of the flank area of abdomen in the next publication.

Osthaus, B., Proops, L., Long, S., Bell, N., Hayday, K., Burden, F., 2018. Hair coat properties of donkeys, mules and horses in a temperate climate. Equine. Vet. J. 50, 339–342. https://doi.org/10.1111/evj.12775.

Comment R.2.8

According to figure 1, the Px1 zone seems solidly defined for its measurement. However, the Px2 area does not seem to have such a standardized determination, since the anatomical structures indicated by the authors in the description are not exactly appreciated in the thermographic image. How did the authors solve this? Do you consider that there could have been less standardization in the measurement of the Px2 area, particularly in terms of the mean temperature?

Answer to R.2.8

Thank you for pointing this out. Yes, we considered the measurement of the Px2 area could have been less standardized. We agree with your opinion and completed the anatomical structures limited the Px2 area. We believe, Area of Tmax, which is computer annotated is much more standardized that manual annotated Px2 area.

Round 2

Reviewer 1 Report

Manuscript ID: animals-1173829-peer-review

Preliminary Study on the Association Between the Area of the Highest Flank Temperature and Concentrations of Reproductive Hormones During Pregnancy in Horses

The authors answered the questions raised by the reviewer. Data of a control group of non-pregnant mares were added. The inadequate sampling frequency for endocrine assessment is mention as a main limitation of the study.

The manuscript falls within the scope of the journal. The current topic is a topic of relevance and general interest to the readers of the journal. Data in Material and Methods should be added (see specific comments). Appropriate statistical analysis is used. The results are discussed and compared with previous publications. However, there are some issues that should be addressed before publication and are summarized below.

Moderate English changes are required.

Title:

Please consider: “Association Between the Area of the Highest Flank Temperature and Concentrations of Reproductive Hormones During Pregnancy in Polish Konik horses- a Preliminary Study”

General comments:

The main purpose of this study corresponds to the use of thermographic imaging as a potentially useful tool in pregnancy detection in native or wildlife horse breeds that cannot be handled. However, the lateral surface of the abdomen must be brushed for dirt and mud removal before imaging (lines 145-146). So, how can this constraint be overcome in order to be used as a diagnostic method of pregnancy in this type of animal?

Specific Comments:

Line 15- please delete ….follow-up….

Line 34- Please considered: …..metabolism intensity. Therefore, this preliminary……

Line 36- please considered: The study was carried out on 14 nonpregnant and 26 pregnant Polish Konik…

Line 37- please delete …….then……

Lines 43-46- Please consider: “Individual thermal characteristics of the lateral surface of the abdomen differed between pregnant and non-pregnant mares in other periods. Differences in maximal and average temperature and Area of Tmax were observed from the 6th month of pregnancy, and those in minimal temperature from the 8th month. “

Lines 114-115- What do the authors mean by this: “Pregnancies were confirmed at 14- and 35-days post-ovulation in 26 mares in the pregnant group and were excluded at corresponding days in 14 mares in the nonpregnant group.”?    Only 12 mares got pregnant?   If the control group corresponds to mares that were not mated (lines 101-103), it does not make sense to say that pregnancy diagnosis was performed in those mares.

Line 122- Was reproductive tract ultrasound examination also performed in control non-pregnant mares during the 11 months of study? If so, please mention it.

Line 124- Were blood samples for reproductive hormone evaluation also performed in control non-pregnant mares during the 11 months of study? If so, please mention it.

Line 143- Were thermographic images collected in control non-pregnant mares during the 11 months of study, on the same day, and in the same environmental conditions of pregnant mares’ group? If so, please mention it.

Line 145- the lateral surface of mares' abdomen was brushed for dirt and mud removal 15 minutes before imaging”. Can this be performed in wild-type horses???

Line 231-232: Please rewrite, since the lowest values occurred not only in the first two months but also in the last two months of pregnancy, with Tamb below 10.0 oC.

Line 237-238- “The highest Tamb (above 20.0oC) was also noted during months 6th and 7th”; this is not correct since, in month 8th, Tamb was higher than in months 6th.

Lines 242-243- Please consider: “The values of Taver Px2, Tmin Px2 decreased to a level higher than in the 1st and 2nd month”.

Lines 314-316- Please rewrite the sentence, since REL concentrations were also high in month 6 of pregnancy (and like those of months 10 and 11) when the lowest area of Tmax was observed.

Lines 324 and 325- please add:  lateral surface of the abdomen

Figures 3 to 9- In order to make it easier for the reader, place above each plot the corresponding thermal feature.

Lines 349 and 361 - please delete mares, since the word is repeated: “….and nonpregnant mares (NP) mares in consecutive …..”

 Lines 351 and 362- In tables 1 and 2, please use different superscripts within the same row to show the statistical differences between months in nonpregnant mares (NP), and use an asterisk to highlight differences between pregnant and nonpregnant groups (* p < .05). In the printed article it is difficult to see the differences marked in bold. Please add the number of mares (n=?) in each group of pregnant and non-pregnant mares.

Line 369- Please correct: non-pregnant

Lines 379- Delete: (2009)

Lines 382- 387: Please consider: “Based on the thermographic imaging of flank area, when pregnant and non-pregnant mares were compared and imaged with stable environmental conditions, it was possible to distinguish between pregnant and non-pregnant mares in the 10th and 11th months of pregnancy [1].

Lines 379- Delete: (2009) and add [1].

Lines 398- Delete: (2009)

Lines 403- 405: Is this what the authors mean?  “Maximum temperature pixel counts seem to be an interesting approach not only because of the lower dependence on ambient temperature but also because of the possible association between Area of Tmax and the concentration of pregnancy hormones in the blood”.

Lines 422- 424: Please consider: “Therefore, this study is a preliminary study to monitor the changes and associations of the thermal characteristics of the lateral surface of the abdomen and the concentrations of reproductive hormones in pregnant Polish native mares.”

Author Response

Dear Reviewer,

We are truly grateful for your opinion that our manuscript falls within the scope of the journal and is a topic of relevance and general interest to the readers. Thank you very much for the substantial amount of time invested in looking over the manuscript, especially the material and methods, statistical analysis, and discussion sections. We have addressed all considerations or explained a few replies. We hope after all changes you have suggested our paper will achieve enough quality to be published in Animals Journal.

Best regards

Comment R.1.1

Please consider: “Association Between the Area of the Highest Flank Temperature and Concentrations of Reproductive Hormones During Pregnancy in Polish Konik horses- a Preliminary Study”

Answer to R.1.1

Thank you for this suggestion. The title has been changed following your comment.

Comment R.1.2

The main purpose of this study corresponds to the use of thermographic imaging as a potentially useful tool in pregnancy detection in native or wildlife horse breeds that cannot be handled. However, the lateral surface of the abdomen must be brushed for dirt and mud removal before imaging (lines 145-146). So, how can this constraint be overcome in order to be used as a diagnostic method of pregnancy in this type of animal?

Answer to R.1.2

Thank you very much for this comment. The brushing of infrared imaging surfaces is one of the parts of the optimal protocol of Infrared thermography in equine medicine (Soroko and Howell, 2018). Without brushing and acclimatization to the optimal imaging conditions, no initial or reference observations may be done. Further research is required to compare brushed and non brushed mares in the same pregnancy state. It is one of the reasons to consider this research as a preliminary study. Following your suggestion, we introduced this information into the discussion section together with other required further measures such as measuring the hair coat lengths and comparison of indoor and outdoor thermographic imaging.

Soroko, M.; Howell, K. Infrared thermography: current applications in equine medicine. J. Equine Vet. Sci. 2018, 60, 90-96. https://doi.org/10.1016/j.jevs.2016.11.002

Comment R.1.3

Line 15- please delete ….follow-up….

Answer to R.1.3

The phrase has been deleted in whole manuscript body.

Comment R.1.4

Line 34- Please considered: …..metabolism intensity. Therefore, this preliminary……

Answer to R.1.4

The sentence has been modified.

Comment R.1.5

Line 36- please considered: The study was carried out on 14 nonpregnant and 26 pregnant Polish Konik…

Answer to R.1.5

The sentence has been modified.

Comment R.1.6

Line 37- please delete …….then……

Answer to R.1.6

Then has been deleted.

Comment R.1.7

Lines 43-46- Please consider: “Individual thermal characteristics of the lateral surface of the abdomen differed between pregnant and non-pregnant mares in other periods. Differences in maximal and average temperature and Area of Tmax were observed from the 6th month of pregnancy, and those in minimal temperature from the 8th month. “

Answer to R.1.7

Thank you for this recommendation, the phrase has been replaced.

Comment R.1.8

Lines 114-115- What do the authors mean by this: “Pregnancies were confirmed at 14- and 35-days post-ovulation in 26 mares in the pregnant group and were excluded at corresponding days in 14 mares in the nonpregnant group.”?    Only 12 mares got pregnant?   If the control group corresponds to mares that were not mated (lines 101-103), it does not make sense to say that pregnancy diagnosis was performed in those mares.

Answer to R.1.8

Thank you very much for point out this insufficiency. To avoid confusion the phrase has been rewritten as following: Pregnancies were confirmed at 14 and 35 days post-ovulation in 26 mares in the pregnant group. In control group, pregnancy were excluded at the beginning of the research in 14 mares. Those mares were introduced into the non-pregnant group.

Comment R.1.9

Line 122-Was reproductive tract ultrasound examination also performed in control non-pregnant mares during the 11 months of study? If so, please mention it.

Answer to R.1.9

Thank you for point it out. Mares were ultrasonographically examined at the beginning of the study. To avoid confusion the phrase has been rewritten.

Comment R.1.10

Line 124- Were blood samples for reproductive hormone evaluation also performed in control non-pregnant mares during the 11 months of study? If so, please mention it.

Answer to R.1.10

Thank you for point it out. Reproductive hormone evaluation were performed in all mares (pregnant and control non-pregnant mares) during the 11 months of study. To avoid confusion the phrase has been rewritten.

Comment R.1.11

Line 143- Were thermographic images collected in control non-pregnant mares during the 11 months of study, on the same day, and in the same environmental conditions of pregnant mares’ group? If so, please mention it.

Answer to R.1.11

Thank you for point it out. Thermographic imaging were performed in all mares (pregnant and control non-pregnant mares) during the 11 months of study. To avoid confusion the phrase has been rewritten.

Comment R.1.12

Line 145- “the lateral surface of mares' abdomen was brushed for dirt and mud removal 15 minutes before imaging”. Can this be performed in wild-type horses???

Answer to R.1.12

Thank you for point it out. We answered this limitation in Answer to R.1.2 and introduced necessary changes into the discussion section.

Comment R.1.13

Line 231-232: Please rewrite, since the lowest values occurred not only in the first two months but also in the last two months of pregnancy, with Tamb below 10.0 oC.

Answer to R.1.13

Thank you very much for this recommendation. The phrase has been replaced.

Comment R.1.14

Line 237-238- “The highest Tamb (above 20.0oC) was also noted during months 6th and 7th”; this is not correct since, in month 8th, Tamb was higher than in months 6th.

Answer to R.1.14

Thank you very much for pointing out this error. It has been corrected.

Comment R.1.15

Lines 242-243- Please consider: “The values of Taver Px2, Tmin Px2 decreased to a level higher than in the 1st and 2nd month”.

Answer to R.1.15

Thank you very much for pointing out this error. It has been corrected.

Comment R.1.16

Lines 314-316- Please rewrite the sentence, since REL concentrations were also high in month 6 of pregnancy (and like those of months 10 and 11) when the lowest area of Tmax was observed.

Answer to R.1.16

Thank you very much for this recommendation. The phrase has been replaced.

Comment R.1.17

Lines 324 and 325- please add:  lateral surface of the abdomen

Answer to R.1.17

Thank you very much for this recommendation. The phrase has been added.

Comment R.1.18

Figures 3 to 9- In order to make it easier for the reader, place above each plot the corresponding thermal feature.

Answer to R.1.18

On each plot the corresponding thermal feature is placed in the left upper corner as the description of each line with square.

Comment R.1.19

 Lines 349 and 361 - please delete mares, since the word is repeated: “….and nonpregnant mares (NP) mares in consecutive …..”

Answer to R.1.19

Thank you very much for this recommendation. The phrases have been deleted.

Comment R.1.20

Lines 351 and 362- In tables 1 and 2, please use different superscripts within the same row to show the statistical differences between months in nonpregnant mares (NP), and use an asterisk to highlight differences between pregnant and nonpregnant groups (* p < .05). In the printed article it is difficult to see the differences marked in bold. Please add the number of mares (n=?) in each group of pregnant and non-pregnant mares.

Answer to R.1.20

Thank you very much for this recommendation. The tables have been corrected.

Comment R.1.21

Line 369- Please correct: non-pregnant

Answer to R.1.21

Thank you for point it out. It has been corrected within the whole manuscript body.

Comment R.1.22

Lines 379- Delete: (2009)

Lines 379- Delete: (2009) and add [1].

Lines 398- Delete: (2009)

Answer to R.1.22

It has been corrected within the whole manuscript body.

Comment R.1.23

Lines 382- 387: Please consider: “Based on the thermographic imaging of flank area, when pregnant and non-pregnant mares were compared and imaged with stable environmental conditions, it was possible to distinguish between pregnant and non-pregnant mares in the 10th and 11th months of pregnancy [1].

Answer to R.1.23

Thank you very much for this recommendation. The phrases have been replaced.

Comment R.1.24

Lines 403- 405: Is this what the authors mean?  “Maximum temperature pixel counts seem to be an interesting approach not only because of the lower dependence on ambient temperature but also because of the possible association between Area of Tmax and the concentration of pregnancy hormones in the blood”.

Answer to R.1.24

Thank you very much for this recommendation. This is exactly what we mean. The phrases have been replaced.

Comment R.1.25

Lines 422- 424: Please consider: “Therefore, this study is a preliminary study to monitor the changes and associations of the thermal characteristics of the lateral surface of the abdomen and the concentrations of reproductive hormones in pregnant Polish native mares.”

Answer to R.1.25

Thank you very much for this recommendation. The phrases have been replaced. 

Reviewer 2 Report

The present manuscript has greatly improved after the first review. Although, in my opinion, it could be published as it is now, I would recommend taking into consideration the following minor suggestions.

ABSTRACT

Abstract. Last sentence. What do the authors want to express with ‘in other periods? I would recommend rewriting this last sentence because it is confusing.

INTRODUCTION

Last sentence. Aim of the study. It would be interesting if the authors include in the aim of the study that changes in thermal features during pregnancy will be compared to those found in non-pregnant mares, subjected to the same conditions.

MATERIAL AND METHODS

Animals.

The authors stated: ‘pregnancies were confirmed at 14- and 35-days post-ovulation in 26 mares in the pregnant group and were excluded at corresponding days in 14 mares in the nonpregnant group’. It is not clear to me if the non-pregnant mares were also subjected to ultrasound examination. As explained by the authors previously, these mares were not mated.

RESULTS

First paragraph. ‘The values of thermal features of the lateral surface of the abdomen different significantly between months of pregnancy’. I would recommend removing ‘of pregnancy’, because these changes have been associated with differences in ambient temperatures, but not with pregnancy. Therefore, the sentence is confusing.

Results. First paragraph. The presentation of results in this first paragraph is quite chaotic. It would be more interesting to present data separated for pregnant and non-pregnant mares or alternatively, data can be described by parameter in each group of mares.

Results. Second paragraph. The authors stated: ‘the pregnancy months with the highest values of measured temperatures corresponded with the highest E1S concentration’. Were these months the hottest? What happened to non-pregnant females? Obviously, there would be no association with hormonal concentration, but it could be the influence of environmental conditions.

Author Response

Dear Reviewer,

We are truly grateful for your opinion that our manuscript has been greatly improved and could be published as it is now. Again thank you very much for your comments which significantly improve the quality of our work. We have addressed them point by point, and hope now our manuscript achieves enough quality to be published in Animals Journal.

Best regards

Comment R.2.1

Abstract. Last sentence. What do the authors want to express with ‘in other periods? I would recommend rewriting this last sentence because it is confusing.

Answer to R.2.1

Thank you for this suggestion. The last sentence of the abstract has been rewritten.

Comment R.2.2

Last sentence. Aim of the study. It would be interesting if the authors include in the aim of the study that changes in thermal features during pregnancy will be compared to those found in non-pregnant mares, subjected to the same conditions.

Answer to R.2.2

Thank you very much for this recommendation. We agree and this phrase has been added to the aim of the study.

Comment R.2.3

The authors stated: ‘pregnancies were confirmed at 14- and 35-days post-ovulation in 26 mares in the pregnant group and were excluded at corresponding days in 14 mares in the nonpregnant group’. It is not clear to me if the non-pregnant mares were also subjected to ultrasound examination. As explained by the authors previously, these mares were not mated.

Answer to R.2.3

Thank you very much for this recommendation. We agree and this phrase has been rewritten.

Comment R.2.4

First paragraph. ‘The values of thermal features of the lateral surface of the abdomen different significantly between months of pregnancy’. I would recommend removing ‘of pregnancy’, because these changes have been associated with differences in ambient temperatures, but not with pregnancy. Therefore, the sentence is confusing.

Answer to R.2.4

Thank you for this suggestion. The sentence has been corrected following your suggestion.

Comment R.2.5

Results. First paragraph. The presentation of results in this first paragraph is quite chaotic. It would be more interesting to present data separated for pregnant and non-pregnant mares or alternatively, data can be described by parameter in each group of mares.

Answer to R.2.5

Thank you very much for this recommendation. We agree the description of the data should be improved, therefore we introduced some changes into the first paragraph of the results section. However, if you agree, we would like to maintained the sequence of data presentation since it is in line with discussion section.

Comment R.2.6

Results. Second paragraph. The authors stated: ‘the pregnancy months with the highest values of measured temperatures corresponded with the highest E1S concentration’. Were these months the hottest? What happened to non-pregnant females? Obviously, there would be no association with hormonal concentration, but it could be the influence of environmental conditions.

Answer to R.2.6

We agree with your opinion and improved this paragraph following your suggestion.